# A Clinical Probability-Based, Stepwise Algorithm for the Diagnosis of Giant Cell Arteritis: Study Protocol and Baseline Characteristics of the First 50 Patients Included in the Prospective Validation Study with Focus on Cranial Symptoms

**DOI:** 10.3390/jcm14072254

**Published:** 2025-03-26

**Authors:** Lukas-Caspar Thielmann, Melike Findik-Kilinc, Louise Füeßl, Christian Lottspeich, Anja Löw, Teresa Henke, Sandra Hasmann, Ilaria Prearo, Amanda von Bismarck, Lilly Undine Reik, Tobias Wirthmiller, Andreas Nützel, Marc J. Mackert, Siegfried Priglinger, Heiko Schulz, Doris Mayr, Elisabeth Haas-Lützenberger, Christina Gebhardt, Hendrik Schulze-Koops, Michael Czihal

**Affiliations:** 1Division of Vascular Medicine, Medical Clinic and Policlinic IV, LMU University Hospital, 80336 Munich, Germany; lukas.thielmann@campus.lmu.de (L.-C.T.); melike.findik@hotmail.de (M.F.-K.); anja.loew@med.uni-muenchen.de (A.L.); teresa.henke@med.uni-muenchen.de (T.H.); sandra.hasmann@med.uni-muenchen.de (S.H.); ilaria.prearo@med.uni-muenchen.de (I.P.); amanda.von_bismarck@med.uni-muenchen.de (A.v.B.); lilly.reik@med.uni-muenchen.de (L.U.R.); tobias.wirthmiller@med.uni-muenchen.de (T.W.);; 2Interdisciplinary Sonography Center, Medical Clinic and Policlinic IV, LMU University Hospital, 80336 Munich, Germany; louise.fueessl@med.uni-muenchen.de (L.F.); christian.lottspeich@med.uni-muenchen.de (C.L.); 3Department of Ophthalmology, LMU University Hospital, 80336 Munich, Germany; marc.mackert@med.uni-muenchen.de (M.J.M.); siegfried.priglinger@med.uni-muenchen.de (S.P.); 4Institute of Pathology, LMU Munich, 80337 Munich, Germany; heiko.schulz@med.uni-muenchen.de (H.S.); doris.mayr@med.uni-muenchen.de (D.M.); 5Department of Hand, Plastic and Aesthetic Surgery, LMU University Hospital, 80336 Munich, Germany; elisabeth.haas@med.uni-muenchen.de; 6Division of Rheumatology and Clinical Immunology, Medical Clinic and Policlinic IV, LMU University Hospital, 80336 Munich, Germany; christina.gebhardt@med.uni-muenchen.de (C.G.); hendrik.schulze-koops@med.uni-muenchen.de (H.S.-K.)

**Keywords:** clinical decision making, clinical prediction rule, diagnostic algorithm, giant cell arteritis, headache, cranial ischemic symptoms, visual loss

## Abstract

**Background:** Early diagnosis of giant cell arteritis (GCA) is crucial to avoid loss of vision, but detailed headache characteristics of GCA have been poorly studied. Clinical prediction rules have shown promise in guiding management decisions in suspected GCA. **Methods:** This is a prospective, monocentric cohort study on patients ≥50 years of age with suspected GCA. The diagnostic efficacy and safety of a previously published prediction rule embedded in a stepwise diagnostic algorithm is compared to the final clinical diagnosis incorporating the results of temporal artery biopsy (TAB). The protocol of the ongoing study is presented in detail. Based on an interim analysis of the first 50 included patients, characteristics of cranial symptoms of patients with positive and negative TAB are compared, and a modification of the original prediction rule is presented. **Results:** TAB was positive in 23 and negative in 26 cases. In one patient, the TAB specimen contained no arterial segment, so this patient was excluded from the interim analysis. Headache was more commonly located temporally and bilaterally. Cranial ischemic symptoms and superficial temporal artery-related symptoms were more common in patients with positive TAB. The quality and intensity of headaches did not differ significantly between groups. As the original prediction rule misclassified a single patient who eventually had a positive TAB, the clinical prediction rule was modified. **Conclusions:** Given the limited sensitivity and specificity of cranial symptoms, a stepwise diagnostic algorithm based on the modified prediction rule may facilitate clinical decision-making in suspected GCA.

## 1. Introduction

Giant cell arteritis (GCA) is the most common primary systemic vasculitis in the elderly. When suffering from the more common cranial manifestation of disease, affected individuals face a substantial risk of ocular ischemic complications, typically occurring before treatment initiation and resulting in permanent visual loss in up to 15% of cases [1]. It is estimated that 500,000 people in Europe, North America, and Oceania will be visually impaired due to GCA by 2050 [2]. Classical symptoms of GCA have been recognized for decades [3], and diagnostic workup of the disease has been substantially improved with the introduction of ultrasound by Schmidt et al., 1997 [4]. Technical developments and methodological innovations, such as the introduction of compression sonography of the cranial arteries nowadays, make high-resolution ultrasound the first-line imaging method in the diagnosis of GCA, with excellent diagnostic accuracy [5]. However, predictive values depend on clinical pre-test probability, which has long been neglected in the diagnosis of GCA. In recent years, several predictive models with varying complexity have been proposed [6,7,8,9,10,11]. Among these is a simple clinical prediction rule published by our group in 2021, which, embedded in a stepwise diagnostic algorithm together with sonographic findings and C-reactive protein (CRP), may facilitate the safe diagnosis or exclusion of GCA [11]. Such an approach may have the potential to avoid overdiagnosis of GCA based on false positive ultrasound findings while ensuring diagnostic safety in patients with suspected GCA.

Here, we describe the study protocol and the baseline characteristics of the first 50 patients included in a prospective validation study of our prediction rule-based diagnostic algorithm, with a focus on detailed characteristics of cranial symptoms and superficial temporal artery physical examination findings. Based on the results of the interim analysis, we present an adaption of the original prediction rule which will be used for final analysis.

## 2. Materials and Methods

### 2.1. Study Design and Study Objectives

The PREDICT-GCA study (clinical prediction rule and high-resolution sonography for the diagnosis of cranial giant cell arteritis: prospective validation study) was approved by the local ethics committee and registered with the German Clinical Trials Register (ID: DRKS00031293). The primary study objective of this ongoing prospective observational study is to determine the negative and positive predictive value of the combination of a simple clinical prediction rule, a sensitive biomarker (CRP), and high-resolution compression sonography of the cranial arteries (hrCS) for the diagnosis or exclusion of cranial giant cell arteritis, compared to sonography only based diagnostic approach [11].

Our study hypothesis states that the misclassification rate (sum of false negative and false positive findings) compared to the final independent clinical reference diagnosis can be significantly reduced by the diagnostic algorithm based on the clinical prediction rule. 

Secondary study objectives are as follows:To determine the diagnostic accuracy of hrCS compared to temporal artery biopsy (TAB) in the diagnosis of cranial GCA.To determine the additional diagnostic benefit of sonographic examination of the facial and axillary arteries in the diagnosis of cranial and extracranial GCA.To determine the benefit of gender- and risk-adapted stratification of the cut-off values of hrCS.To determine the influence of clinical pre-test probability on the diagnostic accuracy of hrCS.To determine the impact of glucocorticoid treatment (dose and duration) on the diagnostic accuracy of the applied test procedures and the diagnostic algorithm.

### 2.2. Clinical Prediction Rule and Diagnostic Algorithm

We published the clinical prediction rule, established on the basis of a derivation cohort and retrospectively validated on a validation cohort in this journal in 2021 [11]. It consists of a demographic criterion (age <70 versus ≥70 years), two typical clinical symptoms (new onset of persistent headache, jaw claudication), and a clinical symptom together with an ophthalmological examination finding (visual disturbance related to anterior ischemic optic neuropathy, AION). Jaw claudication and bilateral AION are each awarded 2 points, whereas the remaining items are scored with 1 point. In both the derivation and the validation cohort, a sum score of <2 points was associated with low rates of GCA as final diagnosis (4% and 1.7%, respectively), whereas a score of ≥2 increased the probability of a final diagnosis of GCA to 70.6% and 51.8%, respectively. Therefore, we apply a cutoff value of 2 points to discriminate patients with low and non-low clinical pre-test probability. The stepwise diagnostic algorithm integrates the clinical pre-test probability (low and non-low), CRP as a sensitive biomarker, and hrCS, as depicted in Figure 1.

Within the algorithm, all patients with non-low clinical probability undergo hrCS, irrespective of CRP values. Patients with low clinical probability are further stratified according to their CRP values. While patients with CRP values ≥ 2.5 mg/dL undergo hrCS, the diagnosis of GCA is rejected without sonographic imaging in patients with CRP values below the cut-off of ≥2.5 mg/dL. In the retrospective validation study (validation cohort), almost half of patients with low clinical probability and CRP values < 2.5 mg/dL would not have been tested with sonographic imaging, and none of these patients had a final clinical diagnosis of GCA [11].

### 2.3. Study Population

Patients ≥50 years of age with a substantiated clinical suspicion of GCA, as determined by experienced rheumatologists, angiologists, or ophthalmologists, are included. The inclusion and exclusion criteria are summarized in Table 1. All patients included agreed to participate in the study and provided written informed consent.

Clinical symptoms are recorded, with a particular emphasis on the duration, character, and intensity of headache. The patients complete a questionnaire with questions on the time of pain onset, the daily course of pain intensity, pain localization, and pain characteristics (Appendix A). Additionally, patients rate their headache intensity and the impact of headache on their quality of life on a 10-point Likert scale.

The clinical examination as part of the study follows a standardized procedure and includes palpation of the temporal arteries (swelling, tenderness, pulselessness) and pulse palpation of the upper extremity arteries as well as axillary artery auscultation. When visual disturbances are present, an ophthalmological assessment including fundoscopy is carried out, and in case of focal neurological symptoms patients are examined by a board-certified neurologist. A specific laboratory marker for the diagnosis of GCA is not available. Laboratory diagnostics include blood count, differential blood count, liver and kidney function tests, thyroid stimulating hormone, standard coagulation parameters, and humoral inflammation markers (CRP and erythrocyte sedimentation rate). Additional laboratory testing, e.g., testing for antineutrophil cytoplasmatic antibodies, is at the discretion of the physicians responsible for patient management.

### 2.4. Sonographic Protocol

Sonography (color duplex sonography and hrCS) is performed in a standardized manner by one of three examiners highly experienced in the diagnosis of GCA and blinded to the clinical information (M.C., C.L., I.P.). To determine interobserver reliability, a subset of patients is examined by two expert sonographers. Sonographic examinations must be performed within 4 days after study inclusion.

All examinations are performed on a LOGIQ E10 machine (General Electric, Milwaukee, WI, USA) with predefined settings. The superficial temporal and facial arteries are visualized with an 18 MHz hockey stick transducer, and for examination of the axillary arteries, a 4–9 MHz linear transducer is used. Wall thickness measures by hrCS sum up the intima-media thickness (IMT) of the near and far wall vessel wall, resulting in values that are approximately twice as high as those specified for classical measurements of the IMT of the far wall in the longitudinal plane [12,13]. Sonographic criteria considered to be diagnostic for GCA are a hypoechogenic wall thickening of at least 0.7 mm on hrCS in at least one temporal or parietal branch of the superficial temporal artery [13] and a noncompressible halo sign of the preauricular main trunk of the superficial temporal artery. A numeric hrCS score is calculated, as described previously [14]. The facial arteries are analyzed exploratively. Circumferential wall thickening of the axillary arteries with thickening of the intima-media complex of the far vessel wall of more than 1.2 mm is diagnostic for extracranial GCA [15].

Sonographic documentation follows a standardized protocol and comprises both representative images and video documentation. All images will be reviewed by an expert panel blinded to the clinical information to establish a final sonographic diagnosis.

### 2.5. Temporal Artery Biopsy and Histology

Unilateral TAB must be performed within 7 days after study inclusion, with the site of biopsy determined according to the sonographic findings. In case of vasculitic wall thickening visualized by hrCS, the superficial temporal artery segment with the most pronounced wall thickening is chosen. In case of unremarkable hrCS, the choice of TAB site is completed by the surgeon performing the procedure. A sample length of at least 1–2 cm is required.

Histological analysis is performed by two pathologists experienced in vascular pathology who are blinded to the clinical information (H.S., D.M.). Processing, interpretation, and reporting of TAB results is conducted in accordance with the consensus recommendations of the Society for Cardiovascular Pathology [16].

### 2.6. Final Reference Diagnosis

The final reference diagnosis will be made by an independent expert panel based on all available data, including the clinical and laboratory findings, imaging results (ultrasound in all patients, additional imaging such as fluorine-18-fluorodeoxyglucose positron emission tomography/computed tomography performed at the discretion of the physicians responsible for patient management), TAB results, and a prospective follow-up of at least 6 months. In the event of disagreement within the panel, all available findings are reassessed, and a consensus decision is made.

Patients with a diagnosis of GCA are treated according to the German guideline recommendations for the management of large vessel vasculitides [17] and are followed up at least every three months according to a fixed protocol. Follow-up care includes routine clinical and laboratory diagnostics as well as sonographic follow-up examinations according to the above-mentioned protocol.

### 2.7. Data Handling

Collected patient-specific data are stored and managed securely in a dedicated data management and analysis system (CentraXX, KAIROS GmbH, Bochum, Germany). Each patient receives an anonymous study identification number under which the clinical data are collected. During the transfer into the database, an automated second pseudonymization ensures safe data handling. Double pseudonymized data are exported for statistical analysis.

### 2.8. Statistical Considerations

The sample size calculation assumes that, based on the primary study hypothesis, the misclassification rate for the final diagnosis of GCA can be reduced from 30% with the purely sonography-based diagnostic approach without clinical risk stratification to 10% with hrCS performed within the diagnostic algorithm. To be able to answer this question in a cohort with approximately one-third of patients with a final diagnosis of GCA, a statistical power (1 − ß) of 80% and an α-error of 5%, 147 cases must be analyzed.

Study objectives addressing diagnostic accuracy of the test strategies are analyzed using 2 × 2 contingency tables to determine sensitivities, specificities, as well as positive and negative predictive values. Parameters measured on a metric scale, e.g., wall thickness in mm, are analyzed by receiver operating characteristic analysis to reassess optimal cut-off values between patients with vasculitis and the ones with alternative diagnoses. For group comparisons, univariate and multivariate significance tests are applied. Statistical analyses are carried out by SPSS 30.0 (IBM Corp., Armonk, NY, USA). Two-sided *p*-values < 0.05 are considered significant.

For this interim analysis, we compared patients with and without positive TAB. Odds ratios were calculated to display the strength of the association between certain clinical variables and a positive TAB. Results for categorical variables are presented as absolute numbers with percentages, and continuous variables are displayed as mean ± standard deviation (SD) or with a 95% confidence interval (CI), as appropriate.

### 2.9. Accompanying Study on Determining Reference Values of hrCS

In previous studies, we have shown that IMT of the cranial arteries depends on age, sex, and cardiovascular risk [14,18]. To define age-, sex-, and cardiovascular risk-adapted reference values for sonography-based arterial wall thickness measurements we initiated an accompanying, already ongoing, prospective monocentric observational study. In a planned number of one thousand patients without previous diagnosis or clinical suspicion of systemic vasculitis or polymyalgia rheumatica, vascular wall structures in the superficial temporal arteries, facial arteries, and carotid arteries are characterized by sonography. Atherosclerotic lesions (plaques) are assessed by quantitative measurements. Questionnaires on the cardiovascular risk profile and physical activity are collected. Cardiovascular risk is estimated by the SCORE2, SCORE2-OP, and SCORE2-Diabetes. After a period of 5 years, a follow-up interview is conducted to assess cardiovascular endpoints (major adverse cardiovascular events). The study was approved by the local ethics committee.

## 3. Results

### 3.1. Baseline Characteristics of the First 50 Patients Included in PREDICT-GCA

Between April 2023 and September 2024, 50 patients were included in the ongoing study, including 27 females and 23 males. Baseline characteristics are listed in Table 2. The mean age of the study participants was 73.9 ± 9.7 years. Nine patients reported a prior history of chronic rheumatic disease. A diagnosis of cardiovascular disease was established in 15 patients, and at least one cardiovascular risk factor was evident in 41 patients. Four patients had a history of malignancy, but none had active cancer.

All patients suffered from cranial symptoms suspicious for a diagnosis of GCA. Any form of headache was present in 74% of patients. Jaw claudication and diplopia were reported in 52% and 10%, respectively. Twenty-six patients experienced persistent visual impairment, which had been preceded by amaurosis fugax in 10 cases and affected one or both eyes in eighteen and eight patients, respectively. Typical symptoms of polymyalgia rheumatica were present in only 10% of patients but 42% reported constitutional symptoms, including fever in four patients.

Antiplatelet therapy, oral anticoagulation, and statin treatment were reported by 11, 8, and 17 patients, respectively. Sixty-eight percent of patients were already under glucocorticoid treatment at the time of the diagnostic workup. Three patients were on long-term low-dose glucocorticoid treatment for pre-existing chronic rheumatic disease, and seventeen patients had already received an intravenous methylprednisolone pulse due to visual impairment. The mean daily dose of prednisolone/methylprednisolone at the time of diagnostic workup was 237 ± 361 mg. The mean time interval between initiation of glucocorticoid treatment and TAB was 3.5 ± 3.3 days.

TAB was diagnostic in 49/50 cases. In one patient, there was no evidence of an arterial vessel by histopathology. This case was excluded from further analysis. TAB was positive in 23/49 patients (46.9%), including 21 patients with classical transmural inflammation and 2 patients with extensive vasculitis of the vasa vasorum and/or periadventitial vessels.

### 3.2. Comparison of Patients with and Without Positive TAB

Headache was reported in three-quarters of patients in both groups (positive TAB: 78.3% vs. negative TAB: 73.1%, *p* = 0.75). Patients with a positive TAB result reported significantly more frequent jaw claudication (78.3% vs. 30.8%, *p* < 0.01), scalp tenderness (56.6% vs. 24.0%, *p* < 0.01), and temporal artery tenderness (69.6% vs. 34.6%, *p* < 0.01). Amaurosis fugax and permanent visual impairment were insignificantly less frequent in the group with positive TAB (amaurosis fugax: 8.7% vs. 30.8%, *p* = 0.08; permanent visual impairment: 43.5% vs. 61.5%, *p* = 0.26). No differences were noted for diplopia (6 vs. 4 patients), polymyalgia rheumatica (3 vs. 2 patients), fever (1 vs. 3 patients), night sweats (9 vs. 12 patients), and extremity claudication (2 vs. 1 patient).

The rate of temporal artery tenderness on palpation did not differ significantly between groups (positive TAB: 56.5% vs. negative TAB: 38.5%, *p* = 0.26). Temporal artery induration (positive TAB: 65.2% vs. negative TAB: 30.8%, *p* = 0.02), as well as temporal artery swelling (positive TAB: 60.9% vs. negative TAB: 15.4%, *p* < 0.01), were significantly more common in the physical examination of patients with positive TAB. Fifteen patients had a normal physical examination, whereas one, two, or three of the above-mentioned findings were present in 14, 12, and 9 patients, respectively. Sensitivity and specificity for prediction of positive TAB were 87%/46.2%, 69.6%/76.9%, and 34.8%/92.3% when at least one, two, or three physical examination findings were present.

Odds ratios of selected symptoms and physical examination findings to predict a positive TAB result are presented in Table 3.

### 3.3. Headache Characteristics

Thirty-six patients (74%) reported new-onset headaches at the time of the diagnostic workup. Approximately one-quarter of patients in both groups reported previous headache episodes (13 of 49 patients in total, all of whom had new-onset headaches that differed in quality and/or intensity from previous headache episodes).

Symptom onset was sudden, happened within a short period of time, or developed gradually over a longer period in 16.7%, 55.6%, and 27.8% of TAB-positive patients and in 33.3%, 44.4%, and 22.2% of TAB-negative patients.

Daily headache symptoms were insignificantly more common in patients with positive TAB (72.2% vs. 50%, *p* = 0.30), while bilateral headache was significantly more common in the positive TAB group (77.8% vs. 42.1%, *p* = 0.04). The detailed headache location in both groups is given in Figure 2A. The mean number of affected regions was 2.7 ± 0.9 vs. 2.2 ± 1.1 in patients with and without positive TAB (*p* = 0.08). Three or more cranial regions were affected in more than half of patients with positive TAB, compared to one-quarter of patients with negative TAB (55.6% vs. 26.3%, *p* = 0.1). Except for temporal headache (73.9% vs. 42.3%, *p* = 0.04), there were no significant differences in headache location between groups. A higher proportion of patients with positive TAB reported neck pain, but this difference was not statistically significant (47.8% vs. 26.9%, *p* = 0.11).

As shown in Figure 2B, thirty-two patients were able to answer specific questions on headache quality (16 patients in each group). Most patients in both the TAB positive and TAB negative groups (12 and 11 patients, respectively), reported dull headaches. In the group of patients with negative TAB, more reported stabbing headaches (five vs. two patients), but this difference was not statistically significant. Other headache qualities were exceptionally rare in both groups.

On a 10-point Likert scale, mean headache severity (positive TAB: 6.3 ± 1.7 vs. negative TAB: 5.6 ± 2.3) and headache-related impairment of quality of life (positive TAB: 5.1 ± 2.8 vs. negative TAB: 4.4 ± 2.8) did not differ significantly between groups (*p* = 0.29 and 0.51).

### 3.4. Clinical Probability Assessment and Amendment of the Clinical Prediction Rule

According to our original prediction rule, the clinical probability was low (0 or 1 point), intermediate (2 or 3 points), and high (≥4 points) in 10, 23, and 16 patients, respectively. The rate of positive TAB was 40%, 26.1%, and 81.3% in patients with low, intermediate, and high clinical probability.

Four out of ten patients with low clinical probability, according to the original prediction rule, had CRP values < 2.5 mg/dL and would not have undergone further diagnostic testing in the original diagnostic algorithm. Three out of these patients had a negative TAB, but one had a positive TAB (with classic transmural inflammation and giant cells). This was a 66-year-old female patient suffering from a new-onset holocephalic headache, massive scalp tenderness, and swollen temporal arteries. As jaw claudication was absent, the patient received just 1 point in the original prediction rule. CRP was 0.7 mg/dL, so no sonographic testing would have been performed within the algorithm.

Acknowledging the failure of the original clinical prediction rule in this single patient, we integrated jaw claudication with other typical cranial ischemic symptoms (scalp tenderness) and temporal artery-related symptoms (temporal artery swelling), which showed high discriminatory values in the above-mentioned analysis. According to the modified clinical prediction rule (Table 4), the clinical probability was low (0 or 1 point), intermediate (2 or 3 points), and high (≥4 points) in 6, 17, and 26 patients, respectively. Of the six patients with low clinical probability, four were men and four were younger than 70 years old, respectively. The rate of positive TAB was 16.7%, 41.2%, and 57.7% in patients with low, intermediate, and high clinical probability according to the modified clinical prediction rule. Three out of six patients with low clinical probability had CRP values < 2.5 mg/dL and would not have undergone further diagnostic testing in the revised diagnostic algorithm. None of these three patients had a positive TAB.

## 4. Discussion

Early diagnosis of GCA is crucial to avoid visual ischemic complications. Ultrasound-based fast-track clinics have shown promise in reducing visual ischemic complications [19,20,21]. However, the diagnostic accuracy of all imaging methods, including ultrasound, depends on pre-test probability. To date, only the 2020 British Society for Rheumatology guideline on the diagnosis and treatment of GCA recommends incorporation of the clinical pre-test probability in the diagnostic workup of suspected GCA [22]. Evidence from prospective studies on the discriminatory value of medical history taking and physical examination in the diagnostic assessment of suspected GCA is scarce.

In this context, we evaluated in detail the clinical characteristics, particularly cranial symptoms and cranial physical examination findings in our interim analysis of the first 50 patients included in a prospective cohort study on the diagnostic accuracy of a stepwise diagnostic algorithm based on a simple clinical prediction rule. We found some meaningful differences between patients with and without positive TAB, with jaw claudication and temporal artery swelling being the variables with the strongest discriminatory value. In 2001, Gonzalez-Gay et al. found a history of constitutional symptoms, an abnormal temporal artery on physical examination, and the presence of visual complications to be the best predictive variables for a positive temporal artery biopsy in a cohort of 190 unselected patients with GCA [23]. In their meta-analysis of 68 studies published in 2020, van der Geest et al. identified jaw claudication and temporal artery abnormalities as clinical variables most informative for a diagnosis of cranial GCA [3]. However, since even these typical symptoms and signs have limited specificity, it is of clinical interest to have a closer look at headache characteristics (i.e., onset, location, quality, and severity of pain) in patients with suspected GCA.

Headache is the most common symptom in patients with suspected GCA, with a prevalence varying between 74% and 87% [6,24,25,26,27], but few studies reported detailed analysis of headache patterns. Back in 1987, a small retrospective study documented that temporal headache occurred in only half of patients with biopsy-proven GCA and exclusively affected the temples in only 25% [28]. Vincenten et al. retrospectively collected anamnestic data from 30 GCA patients and found continuous, unilateral, temporal headaches in 50% of patients [25]. In a prospective study comprising 261 patients with suspected GCA, Toren and coworkers showed that neither headache quality nor location was predictive of a positive TAB [29]. In a prospective cohort study of 56 patients with suspected GCA, Moudrous et al. found that the location of headache did not discriminate between patients with positive vs. negative biopsy [27]. In contrast to Vincenten et al., bilateral headache was more common in patients with positive TAB in the latter study. Sebastian et al. differentiated between generalized and temporal headaches. Both symptoms did not significantly differ in frequency in patients with and without a final diagnosis of GCA in the above-mentioned prospective cohort study [6].

Course and quality of headaches have been poorly studied. Solomon and Cappa reported in their above-mentioned publication that most patients described their pain as throbbing, sharp, dull, or burning and that pain intensity was more often severe than moderate or mild. Continuous and intermittent pain were equally common [28]. Toren et al. reported in their much larger cohort study constant, throbbing, or sharp pain in 36.0%, 19.1%, and 17.5% of cases, respectively [29]. Shimohama et al. described persistent and intermittent pain in 37.5% and 62.5% of patients, and one-third of patients experienced pulsating pain and severe pain intensity, respectively [30]. Our preliminary study adds to the body of knowledge as it shows that temporal and frontal headaches, as well as neck pain, are more common in patients with a positive than a negative TAB. However, headaches affecting at least three regions increased the odds of a positive TAB only insignificantly. Our patients with suspected GCA most frequently reported dull pain, but headache quality did not allow discrimination between patients with and without positive TAB. When interpreting our study results within the context of the available literature, we find strong arguments for using structured approaches to clinical decision-making based on clinical probability assessment.

The original American College of Rheumatology (ACR)-classification criteria from 1990 included new-onset localized headache, temporal artery tenderness, and decreased temporal artery pulse as clinical variables for classification of GCA [31], whereas the 2022 ACR/European League Against Rheumatism (EULAR)-classification criteria included new temporal headache, scalp tenderness, and jaw claudication as three out of six clinical variables [32]. However, classification criteria have not been sufficiently validated for diagnostic purposes. This is one reason why clinical prediction rules for GCA have gained increasing interest in recent years, with some very different approaches published so far. To date, the best-evaluated clinical prediction rule is the Southend pretest probability score (SGCAPS) which, originally published in 2017, assesses GCA probability based on a multiparametric approach including anamnestic information, physical examination findings, and laboratory values [8]. Recently, Sebastian et al. validated the SGCAPS together with sequentially applied quantitative ultrasound in a European prospective multicentric study and proved a high diagnostic accuracy of this combined approach [6]. The authors provided a nomogram that may facilitate clinical probability assessment (low vs. intermediate vs. high, given in percent). Noteworthy, TAB was performed in only 5% of included patients, but the final clinical diagnosis was reassessed during 6 months of follow-up. Canadian neuroophthalmologists proposed a predictive model comprising demographic variables, symptoms, and laboratory values. Their model had similarly good diagnostic accuracy and was retrospectively validated against TAB in 1201 patients with suspected GCA. The authors provided an online risk calculator that estimates clinical probability, also given in percent [9]. Several other clinical prediction rules have been published, some of which were recently externally validated together with the above-mentioned online risk calculator in a retrospective study, with promising diagnostic accuracies [33].

A prospective study compared several probability scores (clinical and sonographic) and showed the highest sensitivity for the SGCAPS. The highest specificity, however, could be reached with another score published as an abstract by Bhavsar-Khalidi in 2019. The authors stated that the combination of a clinical and a sonographic score provides the most accurate diagnostic strategy for suspected GCA [26]. Another strategy is to combine different imaging modalities to increase the diagnostic yield. In a small single-center study from Lecler et al., a diagnostic sequence with high-resolution magnetic resonance imaging followed by temporal artery ultrasound showed a sensitivity and specificity of 100% [34].

In our opinion, the main limitation of the above-mentioned scoring systems is the lack of embedment within a structured diagnostic algorithm. In order to make a clinical decision for an individual patient, the calculated clinical probability must be transferred into a binary diagnostic decision (GCA ruled in or ruled out). In a patient with a calculated high clinical probability of 90%, there is still a 10% chance of an alternative diagnosis, and vice versa. It is of importance in this context that particularly in the setting of optic nerve or retinal ischemia, ultrasound may reveal false positive results in a considerable number of patients [14]. Thus, an ideal clinical decision tool should not only sensitively detect all true-positive cases but also aim at avoiding over-/misdiagnosis (false-positive cases). In other scenarios, clinical decision rules embedded in diagnostic algorithms have been shown to have excellent diagnostic sensitivity but also excellent safety. The best-known diagnostic algorithm, proven to be effective and safe in various prospective management studies, is the Wells-score-based algorithm for deep venous thrombosis and pulmonary embolism [35]. We adapted a similar strategy in our diagnostic algorithm for GCA, with a simple clinical prediction rule, a sensitive biomarker (CRP), and a simple ultrasound method. In the retrospective validation cohort, the approach showed promise in safely avoiding ultrasound studies in patients with low clinical probability while sensitivity for detection of GCA was retained [11].

In contrast to the SGCAPS, our score solely focuses on cranial GCA. Accordingly, the rate of patients with visual impairment was higher in our study than in other cohorts. As the cranial disease pattern is the one putting patients at risk for early ischemic complications and the clinical manifestations of isolated extracranial GCA are highly variable, we believe that a simple approach focusing on cranial GCA is more effective and has a higher chance for broad use not only by vasculitis specialists. Based on the available evidence, we decided to incorporate CRP rather than the erythrocyte sedimentation rate in our algorithm. CRP is slightly more sensitive in the diagnosis of suspected GCA, and the specificity of both markers is low. We applied a CRP cut-off value of 2.5 mg/dL, as the available literature suggests that this cut-off value offers the highest discriminatory value in the diagnostic workup of suspected GCA [3,36].

One single patient categorized as low clinical probability would not have undergone sonographic testing but had positive TAB in the present study. Therefore, we decided to adapt the clinical prediction rule as a consequence of this interim analysis. Detailed prospective data collection allowed for a more precise clinical characterization of patients and scalp tenderness, as well as temporal artery swelling, showing high odds ratios for a positive TAB. The modified clinical prediction rule now includes both parameters. Analysis of the cohort based on the modified clinical prediction rule showed the diagnostic algorithm to be safe. Therefore, the modified algorithm will be used for the final analysis in comparison to the final reference diagnosis. However, only 20% and 12% of patients had low clinical probability according to the original or the revised clinical prediction rule in the present study. The rate was considerably lower than that seen in the retrospective derivation cohort (60.9%) and the validation cohort (52.6%) [11], probably related to a selection bias (investigators may have been reluctant to include patients with low pre-test probability). As the study progresses, we will aim to include more patients with low clinical probability as long as there is a well-founded suspicion of GCA in the individual cases.

Although we routinely assess the axillary arteries by sonography, as suggested by current recommendations, our study did not include cross-sectional imaging of the aorta and its extracranial branches. The application of our algorithm will not exempt physicians from investigating typical symptoms, physical signs, and imaging findings of extracranial GCA. We are aware of the disease patterns and potential complications of extracranial GCA, as exemplified by several publications on this disease pattern by our group [15,37,38,39,40]. However, extracranial GCA as a cause of acute ischemic complications before treatment onset is exceptionally rare.

Our main intention with the modified clinical prediction rule embedded in a diagnostic algorithm is to reduce cranial (visual) ischemic complications due to delayed diagnosis of GCA and to avoid overdiagnosis based on false positive ultrasound results. The interim analysis of our ongoing prospective validation study outlined here in detail is limited by a small sample size. If the efficacy and safety are proven after the inclusion of the 147 planned patients with the assignment of an individual clinical reference diagnosis, our study could build the scientific basis for a prospective multicenter study. 

## 5. Conclusions

Our simple stepwise approach, in contrast to the rather complex scores mentioned above, is intended for broad use in primary health care. It provides a clinical decision-making tool allowing rational initial assessment of clinical probability by physicians not highly experienced in GCA. By restricting referrals to specialized ultrasound fast-track clinics to patients with non-low clinical probability and patients with low clinical probability but elevated CRP, our approach could be effective in reducing costs resulting from the utilization of secondary care.

## Figures and Tables

**Figure 1 jcm-14-02254-f001:**
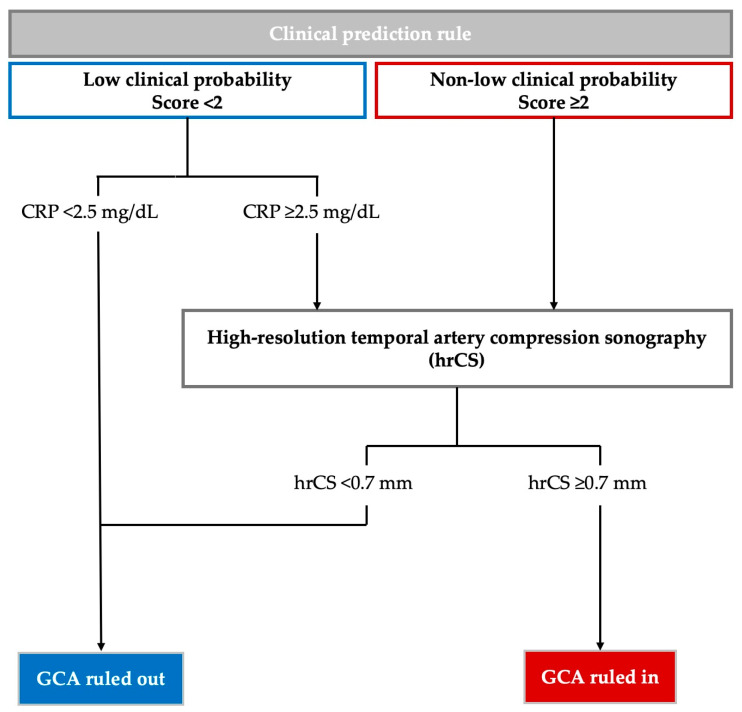
Integrated, stepwise clinical prediction rule for the diagnosis of cranial GCA.

**Figure 2 jcm-14-02254-f002:**
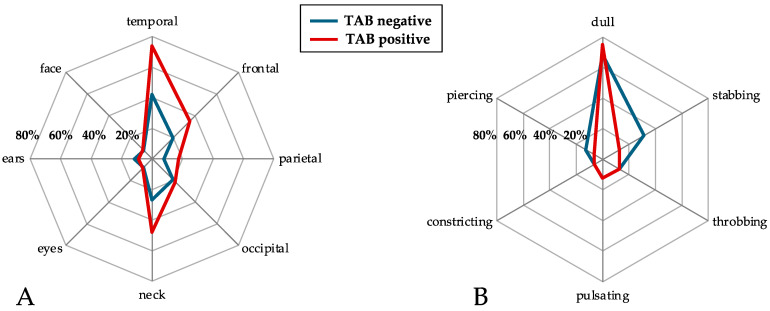
Spider chart depicting the affection of eight head regions by headache (**A**) and six different qualities of headache (**B**), given in percentages of patients.

**Table 1 jcm-14-02254-t001:** Inclusion and exclusion criteria of the PREDICT-GCA study.

Inclusion Criteria	Exclusion Criteria
Age > 50 years	Age < 50 years
Substantiated clinical suspicion of GCA	Pre-existing diagnosis of GCA
Signed informed consent	Glucocorticoid treatment with a daily dose of at least 20 mg prednisolone for >7 days
	Previous glucocorticoid treatment with a daily dose of >20 mg for at least 30 days within the preceding 3 months
	Previous therapy with the interleukin-6 receptor antagonist tocilizumab

**Table 2 jcm-14-02254-t002:** Baseline characteristics of the interim study population.

Variables	Overall Cohort, *n* = 50
Female sex	27 (54)
Age ≥ 70 years	32 (64)
New onset headache	37 (74)
Jaw claudication	26 (52)
Scalp tenderness	19 (38)
Temporal artery tenderness	25 (50)
Diplopia	10 (20)
Amaurosis fugax	10 (20)
Permanent visual impairment	26 (52)
Bilateral visual impairment	8 (16)
Anterior ischemic optic neuropathy	18 (36)
Polymyalgia rheumatica	5 (10)
Constitutional symptoms	21 (42)
Extremity claudication	3 (6)
Manifest cardiovascular disease ^1^	15 (30)
Arterial hypertension	26 (52)
Diabetes mellitus	11 (22)
Active or former smoking	24 (48)
Dyslipidemia	23 (46)
Known rheumatic disease ^2^	9 (18)
History of cancer	4 (8)
Temporal artery swelling	18 (36)
Temporal artery tenderness on palpation	24 (48)
Temporal artery induration	23 (46)
Pre-existing low-dose glucocorticoid treatment	4 (8)
Pre-existing treatment with glucocorticoid sparing agents	3 (6)
Glucocorticoid treatment started for suspected GCA	34 (68)
Daily (methyl-)prednisolone dose (mg)	237 ± 361
Time interval between start of glucocorticoid treatment and TAB (days)	3.5 ± 3.3
C-reactive protein (mg/dL)	4.3 ± 5.7
C-reactive protein elevated	
>0.5 mg/dL	37 (74)
>2.5 mg/dL	21 (42)

Values are mean ± SD or in *n* (%). ^1^ Including coronary artery disease with or without prior myocardial infarction, cerebrovascular disease with or without a history of stroke, peripheral arterial disease, venous thromboembolism, atrial fibrillation, and chronic heart failure. ^2^ Excluding polymyalgia rheumatica or systemic vasculitis.

**Table 3 jcm-14-02254-t003:** Odds ratios of selected cranial symptoms and physical examination findings for prediction of positive TAB in the interim study population.

Variable	Odds Ratio (95% CI)	*p*-Value
Cranial symptoms		
Bilateral headache	3.5 (1.1–11.4)	0.05
Headache in >3 regions	2.0 (0.5–7.1)	0.35
Jaw claudication	8.1 (2.2–29.6)	<0.01
Scalp tenderness	4.1 (1.2–14.2)	0.04
Temporal artery tenderness	4.3 (1.3–14.3)	0.04
Physical examination findings		
Temporal artery tenderness (on palpation)	2.1 (0.66–6.5)	0.26
Temporal artery induration	4.2 (1.3–14.0)	0.02
Temporal artery swelling	8.5 (2.2–33.2)	<0.01

**Table 4 jcm-14-02254-t004:** Items of the modified clinical prediction rule and their relative weightings to create a sum score for dichotomization of clinical pre-test probability.

Variable	Description	Score
Age (years)	<70 years	0
	≥70 years	1
New onset headache	No	0
	Yes	1
Symptoms of cranial ischemia (jaw claudication, scalp tenderness) and/or temporal artery swelling ^1^	No	0
	Yes	2
Permanent visual impairment due to anterior ischemic optic neuropathy	No	0
	Unilateral	1
	Bilateral	2
Score (range 0–6)	**Low clinical** **probability**	**Score < 2**
	**Non-low clinical probability**	**Score ≥ 2**

^1^ Variable modified after interim analysis of the first 50 patients included.

## Data Availability

The data that support the findings of this study are available from the corresponding authors upon reasonable request.

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
