# Peer review of "A Clinical Probability-Based, Stepwise Algorithm for the Diagnosis of Giant Cell Arteritis: Study Protocol and Baseline Characteristics of the First 50 Patients Included in the Prospective Validation Study with Focus on Cranial Symptoms"

_jcm, 2025, doi:10.3390/jcm14072254_

Round 1

Reviewer 1 Report

Comments and Suggestions for Authors

The paper is properly designed and can be useful for standardizing the diagnostic process to highlight this type of vasculitis. The authors should implement clinical stratification by age and sex, furthermore, a clinical evaluation by an ophthalmologist and a neurologist would be useful.

Author Response

Dear Editors,

we appreciate the comments and criticisms of the reviewers. Please find our point-by-point response below. We revised the manuscript accordingly.

Reviewer 1

1.1. The paper is properly designed and can be useful for standardizing the diagnostic process to highlight this type of vasculitis. The authors should implement clinical stratification by age and sex.

Response: Sensitivity analyses stratified by age and sex are planned for the final analysis, but are not feasible in the present interim analysis due to the limited sample size. We have added the information on how many patients with low clinical probability according to the revised prediction rule were male and were under 70 years of age to better characterize this most important subgroup.

1.2. Furthermore, a clinical evaluation by an ophthalmologist and a neurologist would be useful.

Response: For practicability reasons we restricted examinations by board-certified ophtalmologists/neurologists to symptomatic patients (as stated in the manuscript). This is clinically reasonable as it is totally unclear how the detection of asymptomatic ocular/neurological involvement would affect management.

Reviewer 2 Report

Comments and Suggestions for Authors

This is an interesting study. I congratulate the authors on this excellent approach. Methods are sound. The study protocol is well established. The results, including Table 4, are very informative. The Discussion section is very good.

I do not have major points of criticism.

I have a few comments and suggestions that by no means reduce the interest of this elegant study:

Regarding headache, I wonder if it might be interesting to analyze data excluding patients who had reported previous headache episodes earlier in life. Alternatively, headache at the time of GCA diagnosis could be included only if it was clearly different from previous episodes

I suggest that the authors discuss a retrospective study that assessed the predictors of biopsy-proven GCA in 190 Caucasian patients. According to this study, the best predictive model for biopsy-proven GCA included a history of constitutional symptoms (OR = 6.1), an abnormal temporal artery on physical examination (OR = 3.2), and the presence of visual complications (OR = 4.9). In contrast, a negative temporal artery biopsy was more commonly found in GCA patients without visual manifestations, a normal temporal artery on examination, or the absence of constitutional symptoms. Jaw claudication was also less common in GCA patients whose temporal artery biopsy yielded negative results (Ref. PMID: 11182025 -DOI: 10.1053/sarh.2001.16650)

Author Response

Dear Editors,

we appreciate the comments and criticisms of the reviewers. Please find our point-by-point response below. We revised the manuscript accordingly.

Reviewer 2

This is an interesting study. I congratulate the authors on this excellent approach. Methods are sound. The study protocol is well established. The results, including Table 4, are very informative. The Discussion section is very good. I do not have major points of criticism.I have a few comments and suggestions that by no means reduce the interest of this elegant study:

2.1. Regarding headache, I wonder if it might be interesting to analyze data excluding patients who had reported previous headache episodes earlier in life. Alternatively, headache at the time of GCA diagnosis could be included only if it was clearly different from previous episodes.

Response: Thank you for this valuable comment. We have clarified this aspect: in all patients who reported previous headaches, the new headaches had a quality and/or intensity that differed from previous episodes.

2.2. I suggest that the authors discuss a retrospective study that assessed the predictors of biopsy-proven GCA in 190 Caucasian patients. According to this study, the best predictive model for biopsy-proven GCA included a history of constitutional symptoms (OR = 6.1), an abnormal temporal artery on physical examination (OR = 3.2), and the presence of visual complications (OR = 4.9). In contrast, a negative temporal artery biopsy was more commonly found in GCA patients without visual manifestations, a normal temporal artery on examination, or the absence of constitutional symptoms. Jaw claudication was also less common in GCA patients whose temporal artery biopsy yielded negative results (Ref. PMID: 11182025 -DOI: 10.1053/sarh.2001.16650)

Response: Done.

Reviewer 3 Report

Comments and Suggestions for Authors

Dear authors,

I have now completed the review of the manuscript titled "A clinical probability-based, stepwise algorithm for the diagnosis of giant cell arteritis: study protocol and baseline characteristics of the first 50 patients included in the prospective validation study with focus on cranial symptoms."

The manuscript is interesting and, in general, fairly well-written. The study addresses an important clinical problem: early and accurate diagnosis of GCA is crucial to prevent irreversible vision loss, and improved diagnostic algorithms could significantly impact patient outcomes. The stepwise approach combining clinical prediction rules with biomarkers and ultrasound is logical and potentially more resource-efficient than current diagnostic practices. The interim analysis provides detailed characterization of headache patterns in GCA patients, which has been understudied previously. The researchers showed appropriate responsiveness by modifying their clinical prediction rule when they identified a misclassified patient in their interim analysis.

However, I still have some suggestions to further improve the quality of the manuscript.

I would like to suggest that the authors address these limitations in the article, either by discussing them in the limitations section or, where feasible, by making the appropriate revisions:

1. The analysis is based on only 49 evaluable patients (with one excluded due to inadequate biopsy), which limits statistical power and increases the risk that findings might be due to chance.

2. Some important recent findings could be stated in introduction. For example, Global estimates on the number of people blind or visually impaired by age-related macular degeneration: a meta-analysis from 2000 to 2020, since research emphasizes preventing vision loss, this article provides important context on visual impairment epidemiology that could strengthen the rationale.

3. The study uses temporal artery biopsy (TAB) as the gold standard, while simultaneously developing criteria that may guide whether to perform this procedure. This creates a potential circularity in the diagnostic approach.

4. Discussion would be extended with latest research, to show readers future research possibilities. For example, advanced brain tumor segmentation models using multiple MRI modalities might offer valuable techniques transferable to GCA diagnosis, particularly regarding sonography and imaging approaches. Additionally, the research on improving brain stroke detection through multi-layer perceptron neural networks with various optimizers could provide useful methodological frameworks for enhancing the diagnostic algorithms being developed in the current study.

Thank you for your valuable contributions to our field of research. I look forward to receiving the revised manuscript.

Author Response

Dear Editors,

we appreciate the comments and criticisms of the reviewers. Please find our point-by-point response below. We revised the manuscript accordingly.

Dear authors,

I have now completed the review of the manuscript titled "A clinical probability-based, stepwise algorithm for the diagnosis of giant cell arteritis: study protocol and baseline characteristics of the first 50 patients included in the prospective validation study with focus on cranial symptoms." The manuscript is interesting and, in general, fairly well-written. The study addresses an important clinical problem: early and accurate diagnosis of GCA is crucial to prevent irreversible vision loss, and improved diagnostic algorithms could significantly impact patient outcomes. The stepwise approach combining clinical prediction rules with biomarkers and ultrasound is logical and potentially more resource-efficient than current diagnostic practices. The interim analysis provides detailed characterization of headache patterns in GCA patients, which has been understudied previously. The researchers showed appropriate responsiveness by modifying their clinical prediction rule when they identified a misclassified patient in their interim analysis. However, I still have some suggestions to further improve the quality of the manuscript. I would like to suggest that the authors address these limitations in the article, either by discussing them in the limitations section or, where feasible, by making the appropriate revisions:

3.1. The analysis is based on only 49 evaluable patients (with one excluded due to inadequate biopsy), which limits statistical power and increases the risk that findings might be due to chance.

Response: We emphasized this important limitation of our interim analysis in the revised manuscript.

3.2. Some important recent findings could be stated in introduction. For example, Global estimates on the number of people blind or visually impaired by age-related macular degeneration: a meta-analysis from 2000 to 2020, since research emphasizes preventing vision loss, this article provides important context on visual impairment epidemiology that could strengthen the rationale.

Response: In our opinion, global estimates on the number of people blind or visually impaired by age-related macular degeneration do not fit well into the introduction of our manuscript. Instead, we added a study projecting Worldwide Disease Burden from Giant Cell Arteritis and their ocular complications by 2050 (PMID 25362658).

3.3. The study uses temporal artery biopsy (TAB) as the gold standard, while simultaneously developing criteria that may guide whether to perform this procedure. This creates a potential circularity in the diagnostic approach.

Response: This is a well-known, longstanding problem in diagnostic GCA-studies which we stronger emphasized in the concluding paragraph of the revised manuscript. The final analysis will include a final reference diagnosis by independent experts in the field, taking the longitudinal follow-up of the patients into account.

3.4. Discussion would be extended with latest research, to show readers future research possibilities. For example, advanced brain tumor segmentation models using multiple MRI modalities might offer valuable techniques transferable to GCA diagnosis, particularly regarding sonography and imaging approaches. Additionally, the research on improving brain stroke detection through multi-layer perceptron neural networks with various optimizers could provide useful methodological frameworks for enhancing the diagnostic algorithms being developed in the current study.

Response: In our opinion, advanced brain tumor segmentation models using multiple MRI modalities do not fit well into the discussion of our manuscript. Instead, we added a reference, highlighting the potential diagnostic value of different imaging sequences (e.g., MRI -> duplex sonography -> retinal angiography) for the diagnosis of GCA (PMID 34663548).

Round 2

Reviewer 3 Report

Comments and Suggestions for Authors

All comments were addressed.